# Contrast-Enhanced Chest Computed Tomography for *In-Breast* Recurrence Detection: Clinical and Imaging Predictors of Visibility

**DOI:** 10.3390/diagnostics15040407

**Published:** 2025-02-07

**Authors:** Ho Min Jang, Kyoungkyg Bae, Tae Young Lee, Soyeoun Lim, Minseo Bang

**Affiliations:** Department of Radiology, Ulsan University Hospital, University of Ulsan College of Medicine, 877 Bangeojinsunhwando-ro, Dong-gu, Ulsan 44033, Republic of Korea; dc6625@naver.com (H.M.J.); bkkdr@uuh.ulsan.kr (K.B.); 0734147@uuh.ulsan.kr (T.Y.L.); soyeoun.lim.xr@uuh.ulsan.kr (S.L.)

**Keywords:** breast cancer, recurrence, chest computed tomography, mastectomy, tumor visibility

## Abstract

**Purpose:** Routine surveillance chest CT is not recommended by current guidelines; however, its use has been increasing with improved accessibility. This study aimed to evaluate the utility of surveillance contrast-enhanced chest computed tomography (CT) in detecting in-breast recurrence among survivors, focusing on imaging and clinicopathological features that enhance tumor visibility. Additionally, this study sought to determine which patient populations may derive benefit from contrast-enhanced chest CT. **Materials and Methods:** A retrospective analysis was conducted on records of patients diagnosed with in-breast recurrence through biopsy during follow-up after breast cancer surgery between January 2016 and August 2022. Patients who underwent contrast-enhanced chest CT within one month of diagnosis were included. Two radiologists reviewed the chest CT scans for breast cancer lesions by consensus, and their findings were validated by two other radiologists blinded to tumor locations. Statistical analyses were performed to evaluate associations among clinicopathological factors, image features, and visibility. **Results:** Eighty-nine recurrent tumors in 85 patients were included. Fifty-eight recurrent tumors were identified by radiologists who were not blinded. The blinded radiologists independently identified 50 and 56 recurrences, with substantial inter-observer agreement (*κ*-value = 0.768, *p* < 0.001). The visible group had a significantly higher rate of invasive ductal carcinoma (IDC) compared to the non-visible group (81.0% vs. 54.8%, *p* = 0.002). Additionally, the visible group exhibited larger tumors than the non-visible group (mean ± SD: 1.9 ± 1.5 cm vs. 1.3 ± 0.6 cm, *p* = 0.018). Tumors located in fatty backgrounds demonstrated significantly greater visibility on chest CT than those in glandular backgrounds (67.2% vs. 16.1%, *p* < 0.001). Recurrent breast cancer was also more frequently visible on chest CT in patients who had undergone mastectomy compared to those who had received breast-conserving surgery (*p* < 0.001). **Conclusions:** Contrast-enhanced chest CT can aid in the detection of in-breast recurrence, particularly in patients who have undergone mastectomy, as a complementary imaging modality. Tumors in fatty backgrounds, large tumors, mass-type tumors, and IDCs are better visible on chest CT.

## 1. Introduction

Breast cancer survival rates following diagnosis have significantly improved due to advances in treatments and early detection. In 2018, the 5-year survival rate for patients with breast cancer was 91% in the United States [1], and in 2020, it was 93.8% in South Korea. However, its recurrence rate ranges from 6% to 20%, and recurrence after 5 years is common [1,2,3,4]. Women with a breast cancer history have an increased risk of developing breast cancer, which can be either a local recurrence or a new primary cancer in the conserved and contralateral breasts. Detecting asymptomatic breast cancer recurrences through clinical screening is associated with a more favorable prognosis than detecting symptomatic disease [5,6]. Many new therapies are available for patients with recurrent breast cancer, which makes early detection and treatment important [7,8].

Chest computed tomography (CT) is widely used for staging, surveillance, and assessing treatment-related complications in breast cancer patients [9,10]. With technological advancements, chest CT is increasingly performed in clinical practice, often replacing chest radiography as a surveillance tool [11]. Although mammography remains the standard surveillance modality and MRI provides high sensitivity, the use of chest CT for post-treatment follow-up has been rising despite not being recommended for routine surveillance by current guidelines. The increased use of chest CT has resulted in increased detection of incidental findings unrelated to the primary diagnostic area. Two large studies reported a frequency of incidental breast lesions on chest CT of 1.1%, with malignancy rates of 0.4% and 0.3% [12,13]. Several studies have shown that CT significantly contributes to asymptomatic breast cancer detection [14,15]. However, in a previous study, radiologists missed 64.3% of incidental breast cancers on chest CT, indicating the need for thorough examination of the breast area during imaging review [16].

Despite the increasing use of chest CT, its role in detecting in-breast recurrence among breast cancer survivors has not been specifically evaluated. Given its capability for comprehensive thoracic imaging, this study hypothesized that contrast-enhanced chest CT, beyond its primary role in pulmonary surveillance, may also serve as a complementary tool for detecting in-breast recurrence. To investigate this, we conducted a retrospective analysis assessing the detection of in-breast recurrence on surveillance contrast-enhanced chest CT. This study further analyzed the imaging and clinicopathologic characteristics of in-breast recurrence and evaluated past clinicopathologic data to determine which patient populations would benefit most from this imaging modality.

## 2. Materials and Methods

### 2.1. Patients

We retrospectively reviewed the medical records of patients diagnosed with asymptomatic in-breast recurrence through biopsy during surveillance follow-up after curative breast cancer surgery between January 2016 and August 2022 at our institution. The study protocol was approved by the Institutional Review Board, with informed consent waived due to its retrospective nature. Patients who underwent contrast-enhanced chest CT within one month of diagnosis were included. Our institution’s surveillance protocol involves annual chest CT for up to five years post-breast cancer surgery. A total of 95 patients with in-breast recurrence who underwent contrast-enhanced chest CT within one month of diagnosis were included. Ten patients were excluded due to axillary lymph node metastasis (*n* = 3), supraclavicular lymph node metastasis (*n* = 1), inadequate breast parenchyma coverage on chest CT (*n* = 4), and excisional biopsy for the current diagnosis (*n* = 2). Recurrent breast cancer was defined as either ipsilateral or contralateral recurrence. Ipsilateral recurrence referred to local tumor recurrence on the same side after curative breast surgery, while metachronous breast cancer on the opposite side after primary cancer treatment was considered contralateral recurrence [17].

### 2.2. Clinicopathologic Data

We reviewed the patients’ clinicopathological data, including age, body mass index, family history, previous surgery type, radiation therapy history, hormone therapy, chemotherapy, pathologic tumor size, pathologic type, histologic grade, lymph node metastasis status, distant metastasis, extensive intraductal component (EIC), lymphovascular invasion, perineural invasion, hormone receptor status, human epidermal growth factor receptor 2 status, and Ki-67 level.

### 2.3. Assessment of Breast Cancer on Chest CT

Breast cancer on chest CT was assessed in two steps. First, two reviewers with 13 and seven years of experience in breast and chest imaging, respectively, reviewed the contrast-enhanced chest CT images by consensus to identify breast cancer lesions. They compared chest CT findings with those from breast imaging examinations, including mammography, ultrasonography, breast magnetic resonance imaging (MRI), and pathological reports. If the focal breast lesions on chest CT matched those identified on other imaging modalities and in the pathological report, the patient was considered to have a recurrent breast cancer detected on the contrast-enhanced chest CT. Second, to validate the detection on chest CT, two other breast-specialized radiologists, with seven and 18 years of experience, who were blinded to tumor locations, independently examined chest CT scans for focal enhancing breast lesions and recorded the locations individually. The reference standard for breast cancer detection was established based on histopathological confirmation, incorporating findings from dedicated breast imaging modalities such as mammography, ultrasonography, and breast MRI.

### 2.4. Imaging Analysis

The mammographic, ultrasonographic, and MRI scans were retrospectively reviewed in consultation between two radiologists, one with seven years and the other with 13 years of experience in breast and chest imaging. Mammographic density, ultrasonographic background echotexture, and MRI background parenchymal enhancement were analyzed according to the Breast Imaging Reporting and Data System lexicon [18]. On mammography, microcalcifications and masses were noted in the areas where the lesion was found on chest CT. Tumor presentation types, such as mass and non-mass, were analyzed via ultrasonography and MRI. MRI analysis was primarily based on a subtracted 1 or 2 min sequence, with diffusion-weighted imaging (DWI) referenced when necessary. Chest CT was performed to evaluate overlying skin thickening, nipple retraction, and tumor background location (glandular or fatty background). The tumor image size recorded the largest measured size among the imaging modalities.

### 2.5. Techniques of Image Studies

#### 2.5.1. Chest CT

Chest CT was performed using various multidetector (MD) CT scanners after contrast material injection. The scanners included 16-channel (SOMATOM Sensation 16; Siemens Healthcare, Erlangen, Germany), 128-channel (SOMATOM Definition Flash; Siemens Healthcare, SOMATOM go.TOP; Siemens Healthcare, iCT; Philips Healthcare (Cleveland, OH, USA), SOMATOM Definition AS+; Siemens Healthcare), and 384-channel MDCT (SOMATOM Force CT; Siemens Healthcare). The scanning field extended from the lung apices to below the diaphragm. Contrast-enhanced images were obtained 60 s after intravenous injection of the iodinated contrast agent. During the included period, various brands of iodinated contrast agents were used, depending on the timing and type of CT scanners.

#### 2.5.2. Mammography

Standard two-view diagnostic mammography was performed in all patients using the Selenia Dimensions machine (Hologic, Bedford, MA, USA).

#### 2.5.3. Ultrasonography

One of the two radiologists, with 13 and 18 years of experience in this field, performed all ultrasonographic examinations using a handheld 7–20 MHz linear-array transducer (iU22, Philips Ultrasound (Bothell, WA, USA); LOGIQ E10s; GE Healthcare (Milwaukee, WI, USA); EPIQ Elite, Philips Healthcare (Bothell, WA, USA).

#### 2.5.4. MRI

Standard breast MRI was performed using either of three 3-T scanners (Magnetom Vida, Siemens Healthcare, Erlangen, Germany; Magnetom Skyra, Siemens Healthcare, Erlangen, Germany; Achieva, Phillips, Best, The Netherlands) equipped with bilateral 16-channel breast array coils. The MRI protocol included the following pulse sequences: an axial T2-weighted sequence, an unenhanced and contrast-enhanced fat-suppressed axial T1-weighted sequence, and an axial delayed contrast-enhanced fat-suppressed T1-weighted sequence for evaluating supraclavicular and axillary lymph nodes. Six dynamic sequences were obtained before and after intravenous injection of the contrast medium (0.1 mmol kg^−1^ gadoterate meglumine [Dotarem, Guerbet, Villepinte, France]). Post-processing manipulation included the production of standard subtraction, reverse subtraction, multiplanar reconstruction, and maximum intensity projection images.

### 2.6. Statistical Analysis

All statistical analyses were performed using SPSS v. 21.0 (IBM Corp., Armonk, NY, USA). Interobserver agreement was assessed using Cohen’s kappa coefficient to evaluate the consistency between the two blinded radiologists. A *p*-value of <0.05 was considered statistically significant. Fisher’s exact test, Pearson’s χ^2^ test, or an independent t-test was performed to evaluate the associations between clinical pathological factors or imaging findings and visibility on chest CT. Significant parameters in univariate analysis were further analyzed using multivariate logistic regression for past clinicopathologic data associated with visibility.

## 3. Results

In total, 89 recurrences were identified in 85 patients, of which 84 were female and one was male. Sixty-one lesions showed ipsilateral breast recurrence, and 28 lesions showed contralateral recurrence. Of the 89 recurrences, contrast-enhanced chest CT detected 58 cases, resulting in a sensitivity of 65.2% (58/89). The detection was determined by the consensus of two radiologists who were aware of tumor locations. Two other readers, blinded to tumor locations, independently reviewed contrast-enhanced chest CT scans and detected 50 and 56 recurrences, respectively, demonstrating sensitivity values of 56.2% (50/89) and 62.9% (56/89), respectively. Inter-observer agreement between the two blinded readers was substantial (*κ*-value = 0.768, *p* < 0.001).

### 3.1. Characteristics of the Visible and Non-Visible Groups

Table 1 presents the clinicopathologic characteristics of the cohort according to visibility. The pathology type significantly differed between the visible and non-visible groups (*p* = 0.002). The visible group showed a higher rate of invasive ductal carcinoma (IDC) than the non-visible group (81%, 47/58 vs. 54.8%, 17/31). The pathologic lesion size was significantly larger in the visible group than in the non-visible group (*p* = 0.018), with mean sizes of 1.9 ± 1.5 and 1.3 ± 0.6 cm, respectively.

Table 2 summarizes imaging findings based on visibility. On mammography, a mass was observed in 73.8% (31/42) in the visible group, compared to 7.7% (2/26) in the non-visible group (*p* = 0.000). On MRI, a mass-type presentation was observed in 78.8% (41/52) in the visible group, compared to 38.9% (7/18) in the non-visible group (*p* = 0.002). Tumor background location on chest CT also significantly differed between the two groups. In the visible group, a fatty background was observed in 67.2% (39/58), compared to a glandular background observed in 16.7% (5/31) in the non-visible group (Figure 1 and Figure 2).

### 3.2. Past Clinicopathologic Data Associated with Visibility

Past clinicopathologic characteristics based on visibility were analyzed using logistic regression to identify which patient population would benefit from chest CT (Table 3). Adjuvant radiation therapy history and operation type showed significant differences between the two groups. In multivariate logistic regression analysis, the operation type remained an independent and significant factor (odds ratio [OR], 4.243; 95% confidence interval [CI], 1.389–12.964; *p* = 0.011). Notably, recurrent breast cancer was more visible on chest CT in the patient group that underwent mastectomy compared to the group that underwent breast-conserving surgery (Figure 3).

## 4. Discussion

Current guidelines do not recommend routine surveillance chest CT for breast cancer survivors. However, due to patients’ fear of recurrence and clinicians’ preference for early detection of disease recurrence, systemic imaging is frequently used [19]. In a previous survey conducted by the Korean Breast Cancer Society, 50% of the respondents indicated that they performed follow-up chest CT more than once annually in the first five years [11].

Occult breast lesions may be incidentally detected during CT. CT has limitations due to radiation exposure; however, it is faster and more comfortable than MRI. CT can be an alternative when MRI is unavailable, such as in cases of claustrophobia or the presence of a cardiac pacemaker [20]. Several studies have investigated breast lesions detected in chest CT. Tumor size on chest CT correlates well with pathologic tumor size in patients with breast cancer. Features such as irregular shape, spiculated margin, and rim enhancement on CT images have been reported as predictive of malignancy [21,22,23]. However, another study reported that there are no definitive criteria to differentiate benign from malignant lesions [24]. A recent study investigated the quantitative enhancement value of chest CT for distinguishing between benign and malignant lesions, with malignant lesions demonstrating higher enhancement levels [25].

Unlike MRI, which provides sequential dynamic enhancement, chest CT typically involves only one phase acquisition, which is available after contrast media administration. Usually, a scan delay of 55–70 s is maintained following contrast administration on chest CT, corresponding to the early dynamic phase on MRI [26]. Inoue et al. evaluated the time-intensity curve of dynamic MD-dedicated breast CT with four acquisitions (pre, 1, 3, and 8 min), indicating that the washout and plateau patterns between 3 and 8 min had a high positive predictive value (93%) for malignancy [23]. Abbreviated MRI, which captures early enhancement within 2 min without delayed imaging, has been used frequently, and its diagnostic ability is not inferior to that of full dynamic MRI [27]. Delayed dynamic enhanced images cannot be obtained from chest CT; however, images obtained only from the early dynamic enhancement phase still play a role in detecting recurrence.

In this study, only 58 out of 89 recurrent tumors were identified, even with knowledge of tumor locations. Without knowing the tumor location, 50 and 56 recurrent tumors were identified with a substantial degree of inter-observer agreement. This study included only patients with confirmed recurrence. Despite all patients having a recurrence, specialized radiologists could not identify the tumor on chest CT in 34% of cases. Therefore, chest CT may serve as a complementary modality for discovering breast cancer recurrence; however, it is not completely reliable for identifying all recurrences.

The pathology and tumor presentation type of recurrent breast cancer on chest CT and MRI differed between the visible and non-visible groups. Contrast enhancement plays a crucial role in breast cancer detection, as breast cancer typically exhibits higher contrast media uptake than normal breast parenchyma due to neovascularization [28]. This increased vascularization occurs as a result of angiogenic factors, primarily vascular endothelial growth factor (VEGF), which promote both the proliferation of pre-existing capillaries and the formation of new blood vessels. Consequently, malignant breast tumors demonstrate earlier and more intense contrast enhancement compared to normal fibroglandular tissue [29,30]. In this study, invasive ductal carcinoma (IDC) was more frequently visible on chest CT than ductal carcinoma in situ (DCIS). This difference is likely due to the lower degree of neovascularization in DCIS, which may contribute to the non-visualization of some DCIS cases [21,31]. Similarly, a study by Schnall et al. [32] found that 16% of DCIS lesions and 3% of invasive carcinomas exhibited no enhancement on MRI. These findings suggest that the degree of vascularization plays a key role in tumor visibility on imaging modalities. Furthermore, the histological grade and association with DCIS components may also help explain why non-mass-type tumors are less visible on chest CT compared to mass-type tumors. Non-mass-type breast cancers are more frequently associated with lower histological grades and a closer relationship with DCIS, which likely results in reduced neovascularization and weaker contrast enhancement, ultimately leading to lower detectability on imaging [33,34]. In contrast, mass-type tumors tend to demonstrate higher histological grades and more pronounced vascular proliferation, contributing to stronger contrast enhancement and improved visibility on chest CT [33,34].

On chest CT, recurrent tumors are more visible in fatty backgrounds than in glandular tissues. Both glandular tissues and masses appear dense on chest CT and can only be detected if masses are enhanced, but the masses located in the fatty background are easier to detect due to the clear contrast with the surrounding fat [35].

Logistic regression was performed to determine the patient group in which recurrent cancer could be detected better on chest CT using previous clinical pathology data. We found that recurrent cancer was more easily detected on chest CT in patients who had previously undergone mastectomy. This finding is likely related to the fact that tumors located in a fatty background are more easily identified, as observed in this study. Previous studies have reported recurrence rates following mastectomy ranging from 3.6% to 7.7% [36,37]. In many cases, mammography is technically challenging, and its sensitivity for detecting recurrent tumors is inferior to that of physical examination, often relying on ultrasonography or MRI [38,39]. A meta-analysis of surveillance imaging post-mastectomy reported that the overall cancer detection rate per 1000 examinations was 1.86 (95% CI, 1.05–3.30) for mammography, 2.66 (95% CI, 1.48–4.76) for ultrasonography, and 5.17 (95% CI, 1.49–17.75) for MRI [40]. However, in clinical settings, the detection rate of nonpalpable cancers in post-mastectomy surveillance was considerably lower than those of other cancers across all imaging modalities. In asymptomatic patients, conventional surveillance imaging analysis should be performed carefully.

This study has several limitations. First, it included data from multiple CT scanners with varying specifications, which may have influenced tumor visibility. Differences in scanner resolution, reconstruction algorithms, and contrast enhancement protocols could have affected lesion detection. However, with continuous advancements in CT technology, the detection rate of breast masses on CT is expected to improve, potentially enhancing its role in breast cancer surveillance. Second, this study did not include negative cases. The participating radiologists were aware that the patients in this study had a history of breast cancer, which may have introduced bias in lesion detection. This preconceived notion could have led to over-detection of breast cancer, as radiologists may have been more inclined to identify subtle findings that might have been overlooked under routine reading conditions. The lack of a control group with negative cases limits our ability to assess the specificity of contrast-enhanced chest CT in distinguishing malignant from benign lesions. Third, this was a single-center study with a relatively small sample size. The number of patients presenting with isolated in-breast recurrence was limited, which may impact the generalizability of our findings. Given this constraint, single-institution studies inherently face challenges in achieving a sufficiently large cohort for robust statistical analysis. Despite these limitations, our findings suggest that chest CT can serve as a valuable supplementary tool for detecting recurrent breast cancer. To validate the clinical utility of chest CT in surveillance, a larger multicenter prospective study is necessary to further assess its diagnostic performance, including sensitivity, specificity, and real-world applicability.

## 5. Conclusions

This study highlights the potential of chest CT as a valuable adjunctive tool for detecting in-breast recurrence, particularly in patients who have undergone mastectomy. We found that in-breast recurrences detected on chest CT were larger in size, often IDC, located in fatty backgrounds, and presented as masses on MRI with contrast-enhanced chest CT. However, caution is needed when interpreting results, considering the potential for over-detection due to the radiologists’ prior knowledge of patients’ cancer history and the limitations of a single-center study with a small sample size. Larger multicenter prospective studies are required to establish the role of chest CT in routine surveillance.

## Figures and Tables

**Figure 1 diagnostics-15-00407-f001:**
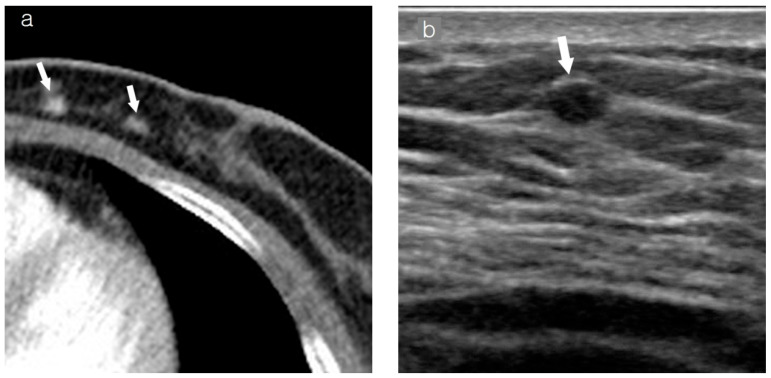
Images of a patient who underwent left breast-conserving surgery 7 years ago. Recently, IDC was diagnosed in the left conserved breast. (**a**) Axial contrast-enhanced chest CT shows an enhancing lesion (arrows) in the inner central portion of the left breast. (**b**) On US, the mass (arrow) appears as a 0.5 cm oval hypoechoic mass located in a fatty background in the left breast. (**c**) CC view of mammography reveals an irregular hyperdense mass (arrow) corresponding to the lesion in the left breast. (**d**) The 2 min subtracted sequence of MRI shows an irregular heterogeneous enhancing lesion (arrows) corresponding to the lesion in the left conserved breast. IDC, invasive ductal carcinoma; CT, computed tomography; US, ultrasonography; CC, craniocaudal; MRI, magnetic resonance imaging.

**Figure 2 diagnostics-15-00407-f002:**
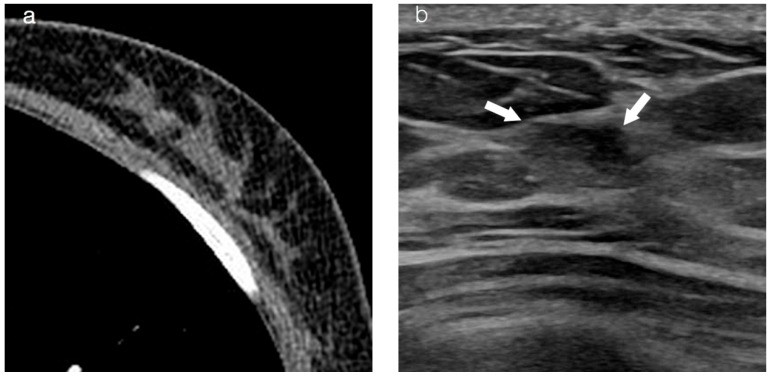
Images of a patient who underwent left breast-conserving surgery 2 years ago. Recently, DCIS was diagnosed in the conserved ipsilateral breast. (**a**) Axial contrast-enhanced chest CT did not show an enhancing lesion in both breasts. (**b**) On US, the mass (arrow) appears as a non-mass-like lesion located in the fibroglandular tissue background of the left breast. (**c**) MLO view of mammography reveals a grouped suspicious microcalcification (arrow) corresponding to the lesion in the left breast. (**d**) The 2 min subtracted sequence of MRI shows a non-mass-like enhancing lesion (arrows) corresponding to the lesion in the left breast. DCIS, ductal carcinoma in situ; MRI, magnetic resonance imaging; CT, computed tomography; US, ultrasonography; MLO, mediolateral oblique.

**Figure 3 diagnostics-15-00407-f003:**
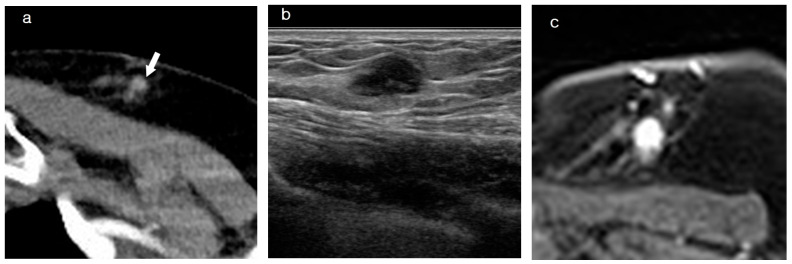
Images of a patient who underwent mastectomy five years ago. (**a**) Axial contrast-enhanced chest CT shows an enhancing lesion (arrow) approximately 1 cm in size at the mastectomy site. Corresponding lesions are also visible on both US (**b**) and 2 min subtracted sequence of MRI (**c**). The discovered lesion was diagnosed as IDC through a biopsy. IDC, invasive ductal carcinoma; MRI, magnetic resonance imaging; CT, computed tomography; US, ultrasonography.

**Table 1 diagnostics-15-00407-t001:** Clinicopathologic factor of patients according to visibility on chest CT.

	Consensus			*p*-Value
	Overall (N = 89)	Non-Visible (N = 31)	Visible (N = 58)	
Age (years)	55.0 ± 10.9	54.0 ± 8.6	55.6 ± 12.0	0.499
BMI (current)	23.8 ± 3.2	23.7 ± 3.7	23.8 ± 2.9	0.963
Menopause (current)				0.925
Pre	31 (34.8)	11 (35.5)	20 (34.5)	
Post	58 (65.2)	20 (64.5)	38 (65.5)	
Family history of breast cancer				1.000
None	79 (88.8)	28 (90.3)	51 (87.9)	
Yes	10 (11.2)	3 (9.7)	7 (12.1)	
Location				0.390
Upper outer	40 (44.9)	10 (32.3)	30 (51.7)	
Upper inner	19 (21.3)	9 (29.0)	10 (17.2)	
Lower outer	15 (16.9)	7 (22.6)	8 (13.8)	
Lower inner	6 (6.7)	2 (6.5)	4 (6.9)	
Subareolar	9 (10.1)	3 (9.7)	6 (10.3)	
Side (current)				0.468
Right	47 (52.8)	18 (58.1)	29 (50.0)	
Left	42 (47.2)	13 (41.9)	29 (50.0)	
Both	0 (0.0)	0 (0.0)	0 (0.0)	
Side (current + past)				0.282
Ipsilateral	61 (68.5)	19 (61.3)	42 (72.4)	
Contralateral	28 (31.5)	12 (38.7)	16 (27.6)	
Pathologic size (cm)	1.7 ± 1.3	1.3 ± 0.6	1.9 ± 1.5	0.018
Estrogen receptor				0.141
Negative	36 (43.4)	9 (32.1)	27 (49.1)	
Positive	47 (56.6)	19 (67.9)	28 (50.9)	
Progesterone receptor				0.240
Negative	46 (55.4)	13 (46.4)	33 (60.0)	
Positive	37 (44.6)	15 (53.6)	22 (40.0)	
Her2				0.103
Negative	43 (51.8)	11 (39.3)	32 (58.2)	
Positive	40 (48.2)	17 (60.7)	23 (41.8)	
Molecular subtype				0.175
Luminal A	24 (28.9)	7 (25.0)	17 (30.9)	
Luminal B	23 (27.7)	12 (42.9)	11 (20.0)	
Her2-enriched	20 (24.1)	5 (17.9)	15 (27.3)	
Triple-negative	16 (19.3)	4 (14.3)	12 (21.8)	
Type				0.002
IDC	64 (71.9)	17 (54.8)	47 (81.0)	
DCIS	14 (15.7)	11 (35.5)	3 (5.2)	
ILC	2 (2.2)	1 (3.2)	1 (1.7)	
Mucinous cancer	2 (2.2)	0 (0.0)	2 (3.4)	
Others	7 (7.9)	2 (6.5)	5 (8.6)	
Lymph node metastasis				1.000
None	66 (84.6)	23 (85.2)	43 (84.3)	
Yes	12 (15.4)	4 (14.8)	8 (15.7)	
Histology grade				0.893
1	3 (5.3)	0 (0.0)	3 (7.0)	
2	30 (52.6)	8 (57.1)	22 (51.2)	
3	24 (42.1)	6 (42.9)	18 (41.9)	
Ki 67				0.492
Ki 67 < 14	24 (30.8)	9 (36.0)	15 (28.3)	
Ki 67 ≥ 14	54 (69.2)	16 (64.0)	38 (71.7)	
EIC				0.334
None	41 (70.7)	9 (60.0)	32 (74.4)	
Yes	17 (29.3)	6 (40.0)	11 (25.6)	
LVI				0.331
None	53 (80.3)	16 (72.7)	37 (84.1)	
Yes	13 (19.7)	6 (27.3)	7 (15.9)	
PN				0.599
None	61 (93.8)	20 (90.9)	41 (95.3)	
Yes	4 (6.2)	2 (9.1)	2 (4.7)	

BMI, body mass index; Her2, human epidermal growth factor receptor 2; IDC, invasive ductal carcinoma; DCIS, ductal carcinoma in situ; ILC, invasive lobular carcinoma; EIC, extensive intraductal component; LVI, lymphovascular invasion; PN, perineural invasion.

**Table 2 diagnostics-15-00407-t002:** Imaging findings of patients according to visibility on chest CT.

	Consensus			*p*-Value
	Overall (N = 89)	Non-Visible (N = 31)	Visible (N = 58)	
**Mammography**				
Calcification				0.259
None	43 (62.3)	14 (53.8)	29 (67.4)	
Yes	26 (37.7)	12 (46.2)	14 (32.6)	
Mass				<0.001
None	35 (51.5)	24 (92.3)	11 (26.2)	
Yes	33 (48.5)	2 (7.7)	31 (73.8)	
Density				0.190
Pattern A	3 (4.3)	0 (0.0)	3 (6.8)	
Pattern B	16 (22.9)	4 (15.4)	12 (27.3)	
Pattern C	41 (58.6)	16 (61.5)	25 (56.8)	
Pattern D	10 (14.3)	6 (23.1)	5 (9.1)	
**Ultrasonography**				
Tumor presentation				0.064
Mass	66 (80.5)	16 (66.7)	50 (86.2)	
Non-mass	16 (19.5)	8 (33.3)	8 (13.8)	
Background echotexture				0.056
Homogeneous fatty	16 (18.6)	2 (6.5)	14 (25.5)	
Homogeneous fibroglandular	41 (47.7)	19 (61.3)	22 (40.0)	
Heterogeneous	29 (33.7)	10 (32.3)	19 (34.5)	
**MRI**				
BPE				0.801
Minimal	53 (74.6)	13 (68.4)	40 (76.9)	
Mild	7 (9.9)	3 (15.8)	4 (7.7)	
Moderate	7 (9.9)	2 (10.5)	5 (9.6)	
Marked	4 (5.6)	1 (5.3)	3 (5.8)	
Tumor presentation				0.002
Mass	48 (68.6)	7 (38.9)	41 (78.8)	
Non-mass	22 (31.4)	11 (61.1)	11 (21.2)	
**Chest CT**				
Background position				<0.001
Fatty portion	44 (49.4)	5 (16.1)	39 (67.2)	
Glandular portion	45 (50.6)	26 (83.9)	19 (32.8)	
Skin thickening				0.125
None	75 (84.3)	29 (93.5)	46 (79.3)	
Yes	14 (15.7)	2 (6.5)	12 (20.7)	
Nipple retraction				1.000
None	83 (93.3)	29 (93.5)	54 (93.1)	
Yes	6 (6.7)	2 (6.5)	4 (6.9)	
Image size (cm)	1.9 ± 1.4	1.6 ± 0.7	2.0 ± 1.6	0.127

BPE, background parenchymal enhancement; **MRI**, magnetic resonance imaging.

**Table 3 diagnostics-15-00407-t003:** Logistic regression analysis of past clinicopathologic factors and visibility on chest CT.

	Univariable			Multivariable		
	OR	95% CI	*p*-Value	OR	95% CI	*p*-Value
Age (current) (years)	1.014	0.974–1.056	0.495			
Age (past) (years)	0.997	0.962–1.034	0.873			
Menopause (past)						
Pre	Ref.					
Post	1.045	0.419–2.605	0.925			
Family history of breast cancer						
None	Ref.					
Yes	1.281	0.307–5.347	0.734			
BMI (past)	1.003	0.874–1.152	0.962			
Adjuvant radiation therapy						
None	Ref.			Ref.		
Yes	0.237	0.088–0.637	0.004	0.474	0.152–1.482	0.199
Adjuvant chemotherapy						
None	Ref.					
Yes	0.721	0.281–1.850	0.496			
Adjuvant hormone therapy						
None	Ref.					
Yes	0.717	0.299–1.722	0.457			
Previous surgery						
Breast-conserving surgery	Ref.			Ref.		
Mastectomy	6.041	2.227–16.384	0.000	4.243	1.389–12.964	0.011
Pathology (past)						
Estrogen receptor						
Negative	Ref.					
Positive	0.665	0.269–1.642	0.373			
Progesterone receptor						
Negative	Ref.					
Positive	0.581	0.237–1.420	0.233			
Her2						
Negative	Ref.					
Positive	0.995	0.411–2.408	0.992			
Molecular subtype						
Lumimal A	Ref.					
Lumimal B	0.485	0.151–1.558	0.224			
Her2-enriched	1.259	0.361–4.391	0.718			
Triple-negative	0.889	0.228–3.459	0.865			
Type						
IDC	Ref.					
DCIS	1.307	0.369–4.628	0.678			
ILC	1.568	0.154–15.936	0.704			
Others	0.174	0.017–1.771	0.140			
Lymph node metastasis						
None	Ref.					
Yes	0.413	0.158–1.081	0.072			
T stage						
0	Ref.					
1	0.630	0.170–2.334	0.490			
2	0.818	0.199–3.369	0.781			
3	0.242	0.029–2.027	0.191			
4	0.364	0.018–7.295	0.508			
N stage						
0	Ref.					
1	0.539	0.148–1.955	0.347			
2	0.449	0.084–2.403	0.349			
3	0.000	0.000.	1.000			

OR, odds ratio; CI, confidence interval; BMI, body mass index; Her2, human epidermal growth factor receptor 2; IDC, invasive ductal carcinoma; DCIS, ductal carcinoma in situ; ILC, invasive lobular carcinoma.

## Data Availability

The datasets used in this study are available upon request from the corresponding author. Access will be provided upon reasonable request and subject to necessary approvals.

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
