# Peer review of "Contrast-Enhanced Chest Computed Tomography for *In-Breast* Recurrence Detection: Clinical and Imaging Predictors of Visibility"

_diagnostics, 2025, doi:10.3390/diagnostics15040407_

Round 1

Reviewer 1 Report

Comments and Suggestions for Authors

<Abstract>

Line 29: The term "effective" in "Contrast-enhanced chest CT is effective for detecting recurrent breast cancer, especially in patients who have undergone mastectomy" is not appropriate. According to the findings of this study, it cannot be regarded as more than a complementary modality. Please revise accordingly.

<Introduction>

Line 64: The term "clinical pathologic data" is used here, but variations such as "clinicopathologic" and "clinicopathological" appear later in the text. Please standardize the terminology to "clinicopathologic" throughout the manuscript.

<Materials and Methods>

Line 88: Revise "(EIC" to "(EIC)".

Line 111: MRI involves numerous sequences. Specify which sequence was used for the analysis.

Lines 118-122: A wide range of CT scanner specifications was used, which could have impacted the results. Add this point to the limitations section.

Line 147: Interobserver agreement is mentioned in the results but is not described in the statistical methods section. Add this information, and if any statistical methods used to derive the results were omitted, provide a detailed explanation.

<Results>

Line 175: Revise "31/58" to "31/42."

Line 180: Revise "531" to "5/31."

<Discussion>

Lines 269-271: The connection between the sentence “In a study by Schnall et al. [32], 16% of DCIS lesions and 3% of invasive carcinomas showed no enhancement on MRI” and the following sentence, “This may also explain why mass-type tumors are more visible on chest CT,” is not smooth. Rewrite it to make the reasoning more coherent and logical. Additionally, the discussion on why mass-type tumors are more visible on chest CT is insufficient. Find and include a more appropriate reference to support this point and revise accordingly.

<Figures>

For the legends of Figures 1-3, provide specific details about the sequences used for chest CT, mammography, and MRI. For example, specify whether it is axial contrast-enhanced chest CT, craniocaudal view vs. mediolateral oblique view for mammography, etc.

Author Response

We sincerely appreciate your time and effort in reviewing our manuscript. Your valuable comments have helped us refine our study and improve the clarity and accuracy of our findings. Below, we provide detailed responses to each of your suggestions and outline the corresponding revisions made to the manuscript. All changes have been highlighted in the revised version for easy reference.

<Abstract>

Line 29: The term "effective" in "Contrast-enhanced chest CT is effective for detecting recurrent breast cancer, especially in patients who have undergone mastectomy" is not appropriate. According to the findings of this study, it cannot be regarded as more than a complementary modality. Please revise accordingly.

  • We appreciate this feedback and have modified the statement to reflect that “Contrast-enhanced chest CT can aid in the detection of in-breast recurrence, particularly in patients who have undergone mastectomy, as a complementary imaging modality”. (p1. Line33-35, p13. Line 339)

<Introduction>

Line 64: The term "clinical pathologic data" is used here, but variations such as "clinicopathologic" and "clinicopathological" appear later in the text. Please standardize the terminology to "clinicopathologic" throughout the manuscript.

  • Thank you for pointing this out. We have stand, arzised terminology and now consistently use 'clinicopathologic' throughout the manuscript. (p2. Line 73 and p9. line 227)

<Materials and Methods>

Line 88: Revise "(EIC를" to "(EIC)".

  • This typographical error has been corrected. (p3. Line 97)

Line 111: MRI involves numerous sequences. Specify which sequence was used for the analysis.

  • We have added details regarding the specific MRI sequences used in the study.(p.3 line 121-123)

Lines 118-122: A wide range of CT scanner specifications was used, which could have impacted the results. Add this point to the limitations section.

  • Thank you for your feedback. We acknowledge the impact that variations in CT scanner specifications could have had on our results. As noted, while the ultrasound (US), MRI, and mammography (MG) devices used in this study had minimal model variation, different chest CT models were utilized. We assessed whether there were differences in visibility based on CT model variation; however, due to the small sample size, the analysis was not statistically meaningful, and no significant differences were observed. Therefore, this analysis was not included in the manuscript.
  • Additionally, as chest CT technology continues to evolve, we anticipate that future clinical outcomes may demonstrate even better results than those presented in our study. To address this point, we have added a statement in the Limitations section (p.13, lines 326-329, 332).

Line 147: Interobserver agreement is mentioned in the results but is not described in the statistical methods section. Add this information, and if any statistical methods used to derive the results were omitted, provide a detailed explanation.

  • We have added details on interobserver agreement analysis in the Statistical Methods section. (p4. line 159-160)

<Results>

Line 175: Revise "31/58" to "31/42."

  • We have corrected this numerical value.(p.6 line 188)

Line 180: Revise "531" to "5/31."

  • We have corrected this numerical error to ensure accuracy.(p.6 line 193)

<Discussion>

Lines 269-271: The connection between the sentence “In a study by Schnall et al. [32], 16% of DCIS lesions and 3% of invasive carcinomas showed no enhancement on MRI” and the following sentence, “This may also explain why mass-type tumors are more visible on chest CT,” is not smooth. Rewrite it to make the reasoning more coherent and logical. Additionally, the discussion on why mass-type tumors are more visible on chest CT is insufficient. Find and include a more appropriate reference to support this point and revise accordingly.

  • Thank you for your valuable suggestion. We have revised the section to improve the logical flow and strengthen the discussion on why mass-type tumors are more visible on chest CT than non-mass-type tumors. The revised text now provides a clearer explanation of the role of vascularization, histological grade, and contrast enhancement in tumor detectability across imaging modalities. (p.12 line 295-306, p15 line 420-421 and 424-425)

<Figures>

For the legends of Figures 1-3, provide specific details about the sequences used for chest CT, mammography, and MRI. For example, specify whether it is axial contrast-enhanced chest CT, craniocaudal view vs. mediolateral oblique view for mammography, etc.

  • We have revised the figure legends to specify the exact imaging sequences used. (p.8 line208-215, p9 line 219-226 and p.11 line 243-245)

Reviewer 2 Report

Comments and Suggestions for Authors

This study aimed to assess the application of contrast-enhanced breast computed tomography (CT) in the detection of recurrent breast cancer among survivors, also to determine which patient populations benefit from this method, focusing on imaging and linear pathological features that enhance tumor visibility.

Although the article contains useful information, there are some ambiguities that need to be clarified, as well as errors that need to be corrected before publication some are listed below:

Abstract

1.       All sentences should be written in a literary form and avoid using the words such as "our article", "our work", "we aimed" etc. instead write "this article aimed to"….

2.       It would be beneficial to mention results statistically in abstract instead of just report the results.

Background

The last paragraph in the introduction section should provide the reader with an overview of the manuscript, but in this article, the abstract and introduction sections do not provide complete information about the method and purpose of the study, it is useful to discuss the use of other methods for detecting recurrence other than CT-scanning to compare the ability of CT while talking about aim of the study.

Results

The use of different methods and different diagnostic devices was excellent and the results are well presented. Did you also examine the differences in the ability of different imaging devices or different contrast materials to detect recurrence?

This study aimed to assess the application of contrast-enhanced breast computed tomography (CT) in the detection of recurrent breast cancer among survivors, also to determine which patient populations benefit from this method, focusing on imaging and linear pathological features that enhance tumor visibility.

Although the article contains useful information, there are some ambiguities that need to be clarified, as well as errors that need to be corrected before publication some are listed below:

Abstract

1.       All sentences should be written in a literary form and avoid using the words such as "our article", "our work", "we aimed" etc. instead write "this article aimed to"….

2.       It would be beneficial to mention results statistically in abstract instead of just report the results.

Background

The last paragraph in the introduction section should provide the reader with an overview of the manuscript, but in this article, the abstract and introduction sections do not provide complete information about the method and purpose of the study, it is useful to discuss the use of other methods for detecting recurrence other than CT-scanning to compare the ability of CT while talking about aim of the study.

Results

The use of different methods and different diagnostic devices was excellent and the results are well presented. Did you also examine the differences in the ability of different imaging devices or different contrast materials to detect recurrence?

Author Response

Reviewer 2

This study aimed to assess the application of contrast-enhanced breast computed tomography (CT) in the detection of recurrent breast cancer among survivors, also to determine which patient populations benefit from this method, focusing on imaging and linear pathological features that enhance tumor visibility.

Although the article contains useful information, there are some ambiguities that need to be clarified, as well as errors that need to be corrected before publication some are listed below:

We sincerely appreciate your time and effort in reviewing our manuscript. Your constructive feedback has been extremely valuable in improving the clarity and quality of our study. Below, we provide a point-by-point response to your comments and describe the corresponding revisions made to the manuscript. All changes have been highlighted in the revised version for easy reference.

Abstract

  1. All sentences should be written in a literary form and avoid using the words such as "our article", "our work", "we aimed" etc. instead write "this article aimed to"….

  • Based on the reviewer's suggestion, first-person expressions have been removed from the abstract and revised to adopt a more academic tone. (p1. Line 11 and 14)

  1. It would be beneficial to mention results statistically in abstract instead of just report the results.
  • We have incorporated key statistical findings into the abstract for greater clarity and specificity. (p.1 line 26-33)

Background

The last paragraph in the introduction section should provide the reader with an overview of the manuscript, but in this article, the abstract and introduction sections do not provide complete information about the method and purpose of the study, it is useful to discuss the use of other methods for detecting recurrence other than CT-scanning to compare the ability of CT while talking about aim of the study.

  • Thank you for your valuable suggestion. We acknowledge that the introduction should provide a more comprehensive overview of the study, including the methodology and study objectives. To address this, we have revised the last paragraph of the introduction to clearly outline the study's aim and methodology while also incorporating a discussion of alternative imaging modalities used for detecting recurrence. (p 2 line 54-59 and 66-74)

Results

The use of different methods and different diagnostic devices was excellent and the results are well presented. Did you also examine the differences in the ability of different imaging devices or different contrast materials to detect recurrence?

  • Thank you for your valuable feedback. We appreciate your recognition of our use of different methods and diagnostic devices.
  • In our study, we primarily focused on the detection of in-breast recurrence using contrast-enhanced chest CT. While we considered variations in CT scanner models, as mentioned in the Limitations section, we did not conduct a detailed analysis comparing the ability of different imaging modalities (e.g., MRI, US, MG) or different contrast materials in detecting recurrence. Given the retrospective nature of our study and variations in clinical practice, such an analysis would require a larger, more controlled dataset. However, we acknowledge that assessing the differences in detection capabilities between imaging modalities and contrast agents would be an important topic for future research.
  • The ultrasound (US), MRI, and mammography (MG) devices used in this study had relatively little variation in models; therefore, we did not perform an analysis based on device differences. However, as various chest CT models were used, we evaluated whether there were differences in visibility based on CT model variation. Although the sample size was small and the analysis may not have been statistically meaningful, no significant differences in visibility were observed among different CT models, and thus, this analysis was not included in the manuscript.
  • Additionally, as chest CT technology continues to advance, we anticipate that future clinical outcomes may yield even better results than those reported in our study. To address this, we have added a statement in the Limitations section (p.13, lines 326-329).

Reviewer 3 Report

Comments and Suggestions for Authors

Dear authors,

   Congratulation for the idea of this paper and for the fact that you decided to study the role of CT scan in the diagnostic of early detection of in breast breast cancer recurrence. However, here are some comments about the paper that I recommend you to take into consideration:

1. Paper is about "in breast " recurrence of breast cancer, NOT about breast cancer recurrence in general. This is not very clear stated nor in title neither in the abstract and main text. Please correct.

2. CT scan as a screening for possible recurrence is not a standard procedure. I read that it is in your institutional rules. Mammography is the only standard radiology  follow up procedure recommended by the guidelines both European and American. This should be very clear stated on the introduction part and in abstract.

3. The association  between in breast recurrence diagnostic  by CT and by mammography and ultrasound and MRI  is important to be added on the objective and results.

4. After "Results", the "Conclusion" section have to be added and to summarize the results.

Comments on the Quality of English Language

Good

Author Response

Reviewer 3

We sincerely appreciate your thorough review of our manuscript and your valuable comments. Your insights have been instrumental in improving the clarity, accuracy, and overall quality of our study. Below, we provide a point-by-point response to each of your comments, along with details of the corresponding revisions made to the manuscript. All changes have been highlighted in the revised version for easy reference.

  1. Paper is about "in breast " recurrence of breast cancer, NOT about breast cancer recurrence in general. This is not very clear stated nor in title neither in the abstract and main text. Please correct.
  • We appreciate this clarification. We have revised the title, abstract, and main text to explicitly state that the study focuses on in-breast recurrence rather than general breast cancer recurrence.(p1 line 2-4, 13,17,34, p2 line 66,70,71, p13 line 340 and 341)

  1. CT scan as a screening for possible recurrence is not a standard procedure. I read that it is in your institutional rules. Mammography is the only standard radiology follow up procedure recommended by the guidelines both European and American. This should be very clear stated on the introduction part and in abstract.
  • We fully acknowledge that chest CT is not a standard surveillance tool and have clarified this point in both the introduction and abstract to reflect the current guidelines. (p1 line 10-11, p2 line 56-59)

  1. The association between in breast recurrence diagnostic by CT and by mammography and ultrasound and MRI is important to be added on the objective and results.
  • Thank you for your insightful comment. We acknowledge the importance of comparing in-breast recurrence detection across different imaging modalities. However, this study primarily focused on analyzing the characteristics of recurrences detected on contrast-enhanced chest CT rather than evaluating the direct correlation between CT, mammography, ultrasound, and MRI. Our objective was to identify the imaging and clinicopathologic features associated with visibility on chest CT, which we believe provides valuable insights into its role as a complementary imaging modality.
  • Nonetheless, we recognize that a comparative analysis of different imaging techniques would further enhance understanding in this field. Due to the retrospective nature of our study and the relatively small sample size, we were unable to perform such an analysis. However, we agree that this is an important area of research and hope to explore this aspect in a well-designed prospective study in the future.
  • We appreciate your suggestion and believe that our findings contribute to the broader discussion of imaging strategies for detecting breast cancer recurrence.
  1. After "Results", the "Conclusion" section have to be added and to summarize the results.
  • The conclusion has been separated into a distinct section to clearly highlight the key findings and implications of the study. (p13, line 338)